# DirectFaaS: A Clean-Slate Network Architecture for Efficient Serverless Chain Communications

Submission Id: 121

## ABSTRACT

Serverless computing, also known as Functions-as-a-Service (FaaS), triggers web applications in the form of function chains. It uses a central orchestrator to route all requests from end-users and internal functions. Such architecture simplifies application deployment for developers. However, the convenient centralized network architecture compromises the efficiency of function chain communications. Specifically, *(i)* a centralized API gateway assists in routing requests between functions. This indirect routing scheme raises invocation latency. *(ii)* The control flow for invoking functions and the data flow for passing function data packets are both forwarded by the API gateway. This results in the API gateway consuming a significant amount of resources. *(iii)* All data packets of internal function communications go through the same API gateway. This expands the additional attack surface in multi-tenant scenarios.

In this paper, we propose DirectFaaS, a clean-slate network architecture to improve the function chain communication performance. By separating coupled control flow and data flow, DirectFaaS releases the API gateway from heavy traffic forwarding, reducing its resource consumption. For this goal, DirectFaaS exploits the network control capabilities of Software-Defined Networking (SDN) to establish direct data forwarding channels to accelerate function chain invocations. In addition, the data flow constrained by fine-grained network policies consolidates multi-tenant traffic security. We implement the DirectFaaS prototype on the popular OpenFaaS platform. Evaluations under real-world serverless applications show that DirectFaaS achieves a reduction in application execution time by up to 30.9% and CPU consumption by up to 30.1% compared to the current architecture.

## KEYWORDS

Serverless computing, Serverless function chain, Serverless networking, SDN

## 1 INTRODUCTION

Serverless computing has gained popularity for deploying web applications [9, 11, 12]. Running web applications without the need to configure the runtime environment is attractive for cloud tenants, especially when their management and provisioning of computing resources are complex. For instance, one in four CloudFront users has embraced serverless computing for frontend development [16].

In serverless computing, web applications run as function chains. Serverless computing platforms, such as AWS Lambda [5], Azure Functions [37], and Google's Cloud Functions [21] facilitate the decomposition of web services [50], Internet of Things [14, 28], machine learning [13, 52], and data analytics [26, 38] applications into Function-as-a-Service (FaaS) and combine functions to form the serverless function chain.

However, function chain communications in the current serverless architecture are not efficient. Serverless function chains are triggered by the API gateway through external events or HTTP requests. Figure 1 shows an example that end-users invoke a function chain with 3 functions. The centralized API gateway assists in forwarding requests between functions, performing 5 internal requests. Compared to direct function-to-function invocations which only need 3 internal invocations, the current architecture introduces 2 additional network round trips. Meanwhile, as cloud applications become more complex, long function chains are quite common. In Azure Durable Functions, 50% of function chains have a length exceeding 3, and even 5% exceeding 8 [35]. The resulting additional network round trips add more execution time for cloud applications, potentially compromising service level objectives (SLOs). The network delays within each region of the serverless platform amplify this function chain invocation latency. As Figure 2 shows, within each region, at least 50% of the delays exceed 2ms, and the latency between regions will be even greater. The function chain invocation latency overheads range from a few to tens of milliseconds [33], making FaaS a poor choice for latency-sensitive interactive applications. Therefore, reducing the latency in the function chain is critical to the performance of serverless computing.

Prior researchers have attempted to reduce the invocation latency in a few different ways. Xanadu [19] and Sequoia [51] proactively warm functions that will execute to reduce function start time. SPRIGHT [42] uses event-based shared memory communication within a serverless function chain to achieve high-speed packet forwarding. QFaaS [22] emerges QUIC protocol to serverless platforms to reduce extra round-trip in TCP. Furthermore, SAND [1] and Nightcore [25] schedule all functions of an application to the same node or the same sandbox to reduce interaction distance, accomplishing low latency.

Nevertheless, all of the current studies still follow the existing network architecture like Figure 3(a) in which the centralized API gateway provides indirect function communications. The centralized network architecture has networking problems from three aspects. First, extra network round-trips are required in function chains, increasing invocation latency (P1). Second, all network flows in function chains are forwarded by a centralized component, prone to a performance bottleneck (P2). Third, multi-tenants share the same centralized component for traffic forwarding, expanding the attack surface of information leakage. (P3).

In this paper, we propose DirectFaaS, a clean-slate network architecture that removes the API gateway from internal function invocations to achieve efficient serverless function chain communications. Specifically, for extra network round-trips (P1), DirectFaaS build direct data forwarding channels between functions to reduce the network round trips in function chain communications. For the centralized component bottleneck (P2), DirectFaaS leverages the SDN's capability to precisely manage network flows. It

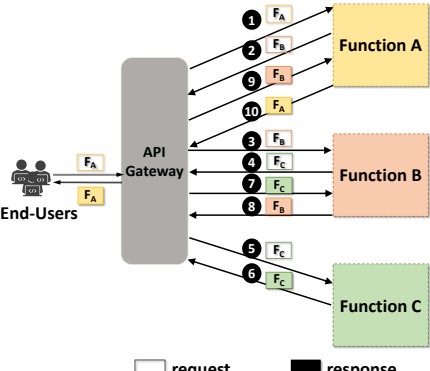
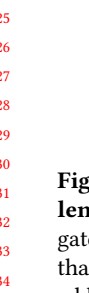
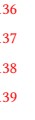

**Figure 1: The process of invoking a function chain with a length of 3.** When end-users invoke the function chain, the API gateway needs to perform 5 request forwarding operations, rather than 3 direct function-to-function invocations. This results in 2 additional network round trips.

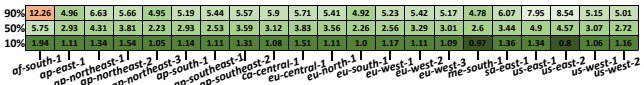

**Figure 2: AWS Lambda intra-region round trip latency (ms) in the 10th, 50th, and 90th percentiles in one year.** Source data is collected from [15] between June 2021 to June 2022.

releases the API gateway from the burden of forwarding internal data flows. For the multi-tenant internal function traffic going through the same component, DIRECTFAAS deploys the network policy [32] that only allows data flows to go through functions with invocation relationships. Fine-grained network policies maintain the isolation of data flows.

The contributions of this paper are as follows:

- We investigate the network architecture of serverless computing and state its networking problems. (§2)
- We present DIRECTFAAS, a clean-slate network architecture. It improves the efficiency of function chain invocations and addresses the challenges of removing the API gateway from internal function invocations. (§3)
- We implement a prototype of DIRECTFAAS on the popular open-source serverless platform OpenFaaS. The entire system code will be made publicly available. (§4)
- We conduct our evaluations of real-world serverless web applications. Compared with OpenFaaS, DIRECTFAAS reduces the execution time of serverless web applications by up to 30.9%. It also reduces the system's CPU usage by 30.1% and memory usage by 13.8%. (§5)

## 2 BACKGROUND AND PROBLEM STATEMENTS

This section introduces the existing network architecture in serverless computing (§2.1) and states networking problems in it (§2.2), which motivate the design of DIRECTFAAS.

## 2.1 Network Architecture in Serverless

In serverless computing, developers only need to upload their code and provide trigger interfaces to form functions. End-users invoke functions in an event-driven way (e.g., HTTP request, timer). Serverless platforms utilize an API gateway to handle function invocations, providing end-users with a simple, flexible, pay-as-you-go way to establish the connection to functions [6]. The API gateway is the piece that ties together serverless functions [46]. It handles all aspects of creating and operating functions for the application. Since functions are designed to be as lightweight as possible to minimize cold start time, they do not have the service mesh to proxy their traffic, all the traffic is routed through the API gateway.

**Function chain invocation workflow.** Figure 3(a) shows an example that end-users invoke an application with two functions. End-users trigger their application through the API gateway (①). The API gateway forwards the request to function A (⑤), accepts invocation requests from function A to function B (⑥), and forwards the requests to function B (⑩). Different serverless platforms may have different approaches to enforcing function chains, but they all have one common feature: *the HTTP requests between functions need to be forwarded by a central orchestrator such as the API gateway.*

**The role of the API gateway.** Since the API gateway handles all aspects of invoking a function, it is not only the data packet forwarding center but also the function control center. The API gateway performs various roles during the function invocation process [6]. It takes responsibility for extensive functionality, including authorization (control flow), scaling functions (control flow), and forwarding application traffic (data flow). As Figure 3(b) shows, the API gateway will authenticate the user's identity, and reject invalid requests(①). Second, when the request is authenticated, the function will be scaled based on the request volume. If a function remains idle for a certain period of time, it will be scaled down to zero. The scaling decision is determined by the invocation count of the function in the API gateway (③). Third, the API gateway forwards requests to the corresponding functions (④). If there are multiple internal invocations, the API gateway scales timely to authorize end-users, resilience functions, and forward traffic.

## 2.2 Networking Problems

The API gateway greatly facilitates function invocations for end-users. However, the effectiveness of serverless networking is compromised for such convenience. Network invocations between function chains are not efficient in current serverless platforms.

**P1: Multiple extra connections increase invocation latency.** A function chain needs to call the API gateway multiple times to forward requests. As Figure 3(a) shows, when end-users invoke a two-function application, 14 steps are involved. There are 4 connections (①, ⑤, ⑥, ⑩) in this function chain. However, it would be more efficient if internal function invocations do not go through the API gateway. If Function A invokes Function B directly, connections (⑥, ⑩) can be avoided.

**P2: The centralized architecture is prone to traffic bottlenecks.** The API gateway performs various roles during the function invocation, including authorization, resiliency, and traffic forwarding.

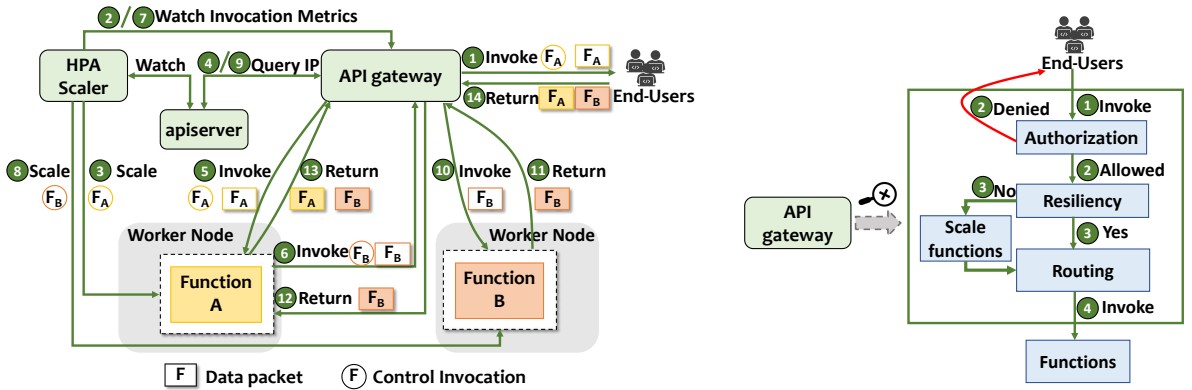

(a) Existing function chain invocation network architecture

(b) The roles of the API gateway in invocation

**Figure 3: The process of invoking the function chain. (a) Network architecture.** End-users invoke the function chain through the API gateway. The API gateway not only forwards requests from end-users but also from Function A. **(b) API gateway invocation processes.** When end-users invoke a function, the API gateway needs to perform 3 roles. Authorization for end-users, scaling functions, and routing requests to functions.

Although the API gateway can scale timely when facing burst traffic, the heavy tasks consume numerous resources in the platform, making the API gateway become the traffic bottleneck.

**P3: Lack of data flow isolation compromises security.** The data flows from multi-tenants coexist within the API gateway. Even though serverless platforms like AWS employ role-based access control policies to maintain tenant's data isolation [7], the internal attack surface is still expanded in the current serverless architecture. For instance, Virtual Private Cloud (VPC) provides an isolated networking environment in public clouds. In serverless computing, customized VPCs are actually disabled in AWS Lambda by default [4]. All Lambda functions from the different tenants actually share the same public VPC. Multi-tenant functions can access the same API gateway.

## 3 SYSTEM ARCHITECTURE DESIGN

In this section, §3.1 introduces challenges in designing ideal network architecture. We propose DIRECTFAAS and discuss how it addresses these challenges in §3.2. §3.3 and §3.4 provide detailed designs of DIRECTFAAS control flow and data flow, respectively.

### 3.1 Challenges

We believe that direct communication between functions is the key to addressing these networking problems (§2.2). To enable direct communications, we need to remove the API gateway from internal function invocations and decouple the data flow from the control flow. However, due to the various roles that the API gateway performs during the function invocation (Figure 3(b)), releasing the API gateway from data forwarding is not easy. DIRECTFAAS's design should tackle three key challenges:

**C1: Handing the authorization of the internal function invocation.** Serverless platforms [5, 21, 37] authorize both external and internal function invocations through the API gateway. For external invocations, the API gateway checks whether users' bearing tokens or request parameters are authorized [10]. For internal invocations, it will also determine whether a function is qualified to

access another function. However, if internal invocations no longer go through the API gateway, it is challenging to validate function chain authorizations.

**C2: Routing functions without knowing the IP address.** In current serverless platforms, the API gateway is responsible for routing requests between functions. When a function invokes another function, technically, it invokes the API gateway with the function name as a parameter. Functions scales a different number of instances based on the traffic volume. The API gateway is responsible for finding the IP address of the invoked function instance and forwarding the request. However, if internal function invocations no longer go through the API gateway, function instance addressing becomes a challenge.

**C3: Scaling functions when functions are scaled to zero.** Using "scale to zero" to achieve "pay-as-you-go" is one of the advantages of serverless computing. When a function is not invoked for a period of time, the number of running function instances will be scaled down to 0 by platforms. If a zero-scaled function is invoked, Horizontal Pod Autoscaling (HPA) Scaler will scale up this function based on invocation metrics in the API gateway [8]. However, if a function directly invokes a zero-scaled function, invocation metrics in the API gateway will not be updated. As a result, the function will not be scaled in the current architecture, which becomes a challenge.

### 3.2 System Architecture Overview

From the insights above, we design DIRECTFAAS, a serverless network architecture that not only improves the function chain invocation efficiency but also solves the challenges discussed in §3.1.

DIRECTFAAS architecture separates the control plane and the data plane in serverless chain communications. Components of the control plane are responsible for generating control flows. Following these control flows, internal functions communicate directly in the data plane. As shown in Figure 4, DIRECTFAAS controller in the control plane manipulates control flows, while Virtual Switches forward data flow in the data plane. The detailed design of DIRECTFAAS controller and Virtual Switch are highlighted below.

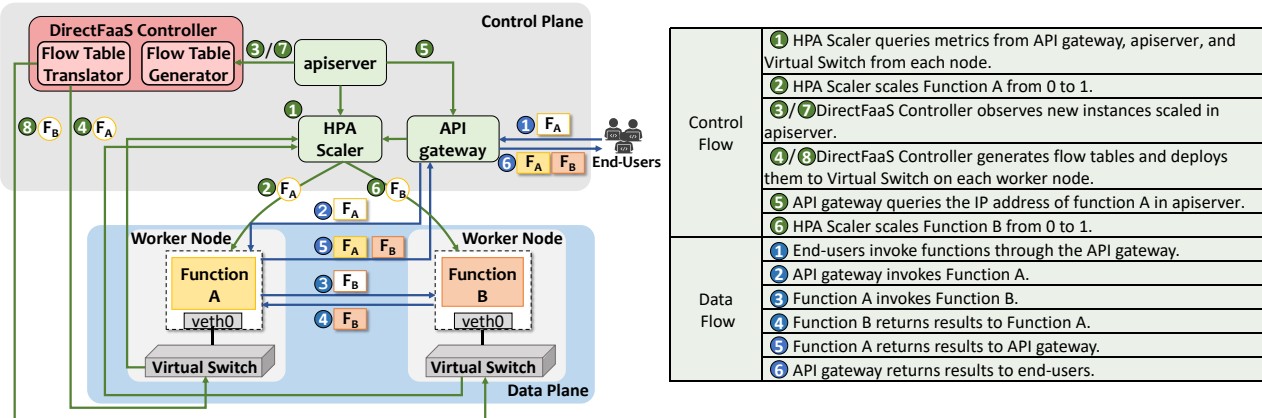

Figure 4: DIRECTFAAS serverless network architecture. The green arrows represent the control flows. The blue arrows represent the data flows. We separate control flows and data flows, making direct communications between Function A and Function B possible.

**DIRECTFAAS Controller.** We introduce DIRECTFAAS controller to the serverless network architecture. It has SDN [36] capabilities that can deploy network policies and allocate static IPs. These capabilities address C1 and C2 in §3.1. Although the DIRECTFAAS controller is also a centralized component, different from traditional API gateway, it only generates and watches control flows.

**Solution to C1: authorizing internal functions through the network policy.** In our architecture, the API gateway still handles authorization and authentication for the invocations from end-users. However, for internal function invocations that do not go through the API gateway, we use SDN capability to achieve network policies to restrict internal invocations. The DIRECTFAAS Controller consists of a *Flow Table Generator* and a *Flow Table Translator*. The *Flow Table Generator* generates the network policy based on the application workflow's Directed Acyclic Graph (DAG). The *Flow Table Translator* deploys them to the data plane. The network policy restricts arbitrary invocations between internal functions, allowing access to functions only if there is an invocation relationship in the DAG. As Figure 5 Flow tables show, if functions have no invocation relationship with Function B, its requests will be denied (Line 1). Function A has an invocation relationship with Function B, so requests from Function A are allowed to reach Function B (Line 2). Even without authentication between internal functions through the API gateway, fine-grained network policies still maintain the security of internal function invocations.

**Solution to C2: using the static virtual IP (vIP) to route internal function communications.** When the API gateway no longer assists with internal addressing and routing, we design the static virtual IP (vIP) for direct routing. Each function comprising the application will be allocated a vIP address that does not change until the function is removed from the platform. Functions can directly invoke each other using the vIP address. When a new function instance is created, the Flow Table Generator will add a flow table rule that maps the vIP to the dynamic endpoint IP (eIP) of this instance. As Figure 5 Flow tables show, when Function A directly invokes the vIP of Function B, as the packet goes through the Virtual Switch, its destination IP will be changed to the eIP of

Function B's instance (Line 3). Therefore, requests from Function A will be forwarded directly to Function B without the need for API gateway addressing and routing.

**Virtual Switch.** Virtual Switches are widely used in current cloud platforms [2]. In DIRECTFAAS design, Virtual Switches are responsible for networking connectivities of function instances. The number of invocations of each function recorded by the Virtual Switch is crucial for addressing the function scaling challenge (C3) in §3.1.

**Solution to C3: monitoring internal function invocations to scale function instances.** When the API gateway no longer updates the number of internal function invocations, we design the HPA Scaler to get metrics of function invocations from the Virtual Switch. Since all the function traffic goes through the Virtual Switch, the Virtual Switch can accurately update the number of invocation times of each function. When Function A invokes Function B which is zero-scaled, the Virtual Switch will update the number of the invocation times of Function B. Thus, the HPA Scaler can scale Function B from 0 to 1, which solves the function scaling challenge.

## 3.3 Control Flow Design

Green arrows in Figure 4 represent the control flows that are sent by control components. They are responsible for monitoring metrics, scaling functions, and generating flow table rules. The control flows c①, c③, c⑤, and c⑦ continuously monitor function metrics. The HPA Scaler gets function invocation counts from the API gateway, Virtual Switches, and gets the function resource utilization from the apiserver (c①). Based on these metrics, the HPA Scaler scales the functions. When a new function instance is created, the DIRECT-FAAS Controller obtains the endpoint IP address of the function instance from the apiserver [30] for generating flow table rules (c③, c⑦). The API gateway obtains the eIP of the ingress function for forwarding end-user requests (c⑤). However, it will no longer address internal invocations, which means that when an end-user invokes a function chain, the API gateway only needs to query the IP address once. The control flows c② and c⑥ scale functions when the HPA Scaler observes the function invocation count transmitted from 0 to 1. The control flows c④ and c⑧ generate flow table

| Flow table | | | |
|---|---|---|---|
| | **Source IP** | **Destination IP** | **Actions** |
| Line 1 | Except for eIPs of Function A | 10.64.0.101 (vIP of Function B) | Deny |
| Line 2 | 10.16.0.200 (eIP of Function A) | 10.64.0.101 (vIP of Function B) | Allow |
| Line 3 | 10.16.0.200 (eIP of Function A) | 10.64.0.101 (vIP of Function B) | Change Destination IP to 10.16.0.1 (eIP of Function B) |

**Figure 5: Data flow between two functions.** When Function A invokes Function B, the action is performed on the packet on the Virtual Switch. According to the flow table, changes the virtual IP (vIP) of the packet to the endpoint IP (eIP), and forwards it to Function B.

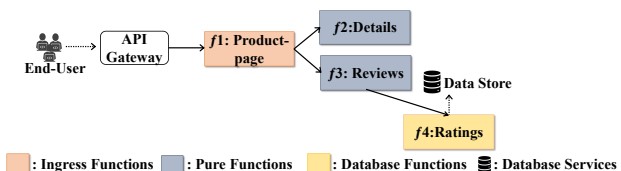

: Ingress Functions    : Pure Functions    : Database Functions    : Database Services

**Figure 6: The architecture of *Bookinfo.***

| Application Functions | Invoked Functions | | | |
|---|---|---|---|---|
| | f1 | f2 | f3 | f4 |
| Product-page (f1) | × | 300/300 | 300/300 | × |
| Details (f2) | × | × | × | × |
| Reviews (f3) | × | × | × | 300/300 |
| Ratings (f4) | × | × | × | × |

**Table 1: The functions in *Bookinfo* invoke each other.** Functions can only be invoked by other functions that have an invocation relationship.

rules and deploy them to Virtual Switches when the DIRECTFAAS Controller gets the new eIP from the apiserver. These control flows have the following properties:

**Programmability.** Programmability is a fundamental feature for the DIRECTFAAS Controller. The control flows generated by the DIRECTFAAS Controller can be programmed by the serverless platform providers.

**Transparency.** All the modifications in DIRECTFAAS are transparent to end-users and existing serverless applications. End-users still interact with the API gateway to invoke function chains. Existing serverless applications can directly deploy on DIRECTFAAS without any modification.

### 3.4 Data Flow Design

The data flow is divided into two parts: the data flow interacts with the API gateway and the data flow between the internal function invocations. In Figure 4, the data flows d① and d⑥ are responsible for the interaction between end-users and the API gateway. End-users invoke functions with parameters (d①) and the API gateway returns results to end-users (d⑥). The data flows d② and d⑤ are responsible for transmitting data between the API gateway and the ingress function, including invoked parameters to the function chain (d②) and the returned computation result from the function chain (d⑤). The data flows d③ and d④ are transmitted directly

between internal function chains. As Figure 5 describes, the Virtual Switch forwards the data packets to the destination function, achieving a direct connection between internal functions. The direct data flows between functions have the following properties:

**Scalability.** In traditional serverless architecture, data flows are routed by the API gateway. While the API gateway can scale, it still becomes a bottleneck for traffic. DIRECTFAAS enables functions to communicate peer-to-peer, eliminating the bottleneck. Data flows exhibit greater scalability.

**Isolation.** The internal data flow no longer goes through the API gateway where multiple tenant traffic is aggregated. The network policy strictly restricts the direction of the internal function data flow to maintain their isolation and security.

## 4 IMPLEMENTATION

We implemented our architecture on OpenFaaS [40]. OpenFaaS is one of the most popular open-source serverless platforms that gets 23.6k starts in GitHub [41]. It is a container-based serverless platform [48] and orchestrated by the Kubernetes [31] infrastructure. We provide detailed implementations of DIRECTFAAS five components. We will open-source our code of the whole system and the way we test the system.

**API gateway.** We use OpenFaaS's API gateway as our API gateway prototype. We maintain the API gateway's role in handling end-user requests. Since internal functions in DIRECTFAAS no longer go through the API gateway, we exclude the API gateway from internal functions forwarding. We deploy network policies for authentication between functions and generate flow table rules for routing internal functions.

**Apiserver.** Since we deployed OpenFaaS over Kubernetes, the Kubernetes API server works as the apiserver. It provides HTTP REST API interfaces to add, delete, update, and watch functions. The API gateway and the HPA Scaler interact with the Kubernetes API server to manipulate the function resource. DIRECTFAAS Controller interacts with the apiserver periodically to watch function instances.

**HPA Scaler.** We use KEDA [27], a Kubernetes-based event-driven autoscaler to scale functions based on the number of function invocations. When the function invokes other functions directly, the Virtual Switch will update the number of invoked functions. Then KEDA observes the change and uses an event-driven way to scale functions.

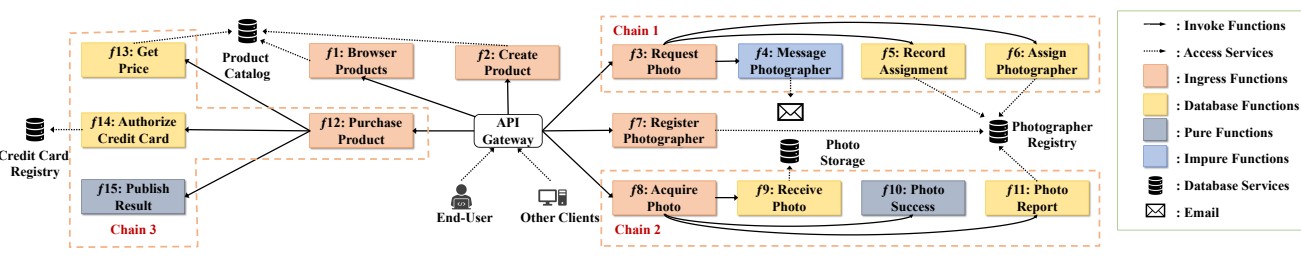

Figure 7: The architecture of *Hello Retail!* Here is annotated to demonstrate three function chains. The orange frame surrounds a function chain and black arrows denote the invocation relationship in the serverless function chain. The ingress functions can be called by end-users by the API gateway but other functions can only be accessed by the ingress functions.

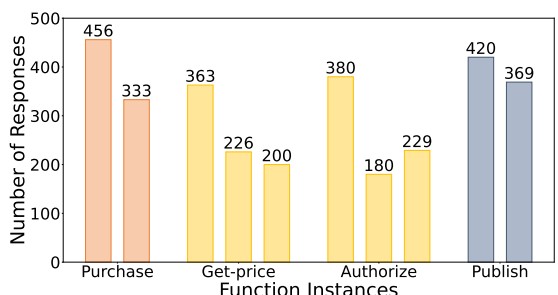

Figure 8: Internal functions are scaled when heavy traffic bursts in *Hello Retail!* function chain 3. In function chain 3, functions scale different numbers of instances. Each function successfully responds to the sent 789 requests.

**DirectFaaS Controller.** The DirectFaaS Controller is implemented based on the kube-OVN controller. Kube-OVN [29] is a cloud-native computing foundation sandbox-level project that integrates the SND-based network virtualization with Kubernetes. We bring the SDN capabilities into serverless platforms. Our modifications are aimed toward allocating the vIP for each function, generating flow tables when there are new functions scaled.

**Virtual Switch.** We configure and operate the network bridge using Open vSwitch (OVS) [39]. It exists on each worker node as a part of the network stack, consisting of the network bridges. When a function is scaled, a port corresponding to the function's veth interface will be created on the OVS. The flow table rules deployed on the Virtual Switch route packets of functions to the correct destination.

## 5 EVALUATION

In this section, we evaluate the functionality and performance of DirectFaaS. We deploy DirectFaaS on a 3-node Kubernetes cluster (v1.23.6). Each node is equipped with 8x 2.20-GHz Intel Xeon CPUs (E5-2650) and 32GB memory running Ubuntu 20.04 TLS. Our experiments use Docker version 20.10.17 as the container runtime and OpenFaaS with gateway version 0.23.0.

**Application workloads.** We use three serverless applications to evaluate the performance of DirectFaaS against OpenFaaS, including *(i)* a web application that displays book information *Bookinfo*, *(ii)* a commercial serverless application *Hello, Retail!*, and *(iii)* a synthetic serverless application with variant function chain lengths.

*Bookinfo* [24] consists of four functions that are written in different languages. The architecture is shown in Figure 6. The Productpage is the ingress function (publicly accessible) that is invoked by end-users through the API gateway. Other functions are internal functions (can only be accessed by ingress functions).

*Hello, Retail!* [49] has been extensively used in recent serverless studies [17, 18, 22, 45]. As shown in Figure 7, *Hello Retail!* consists of 15 functions, 6 of which are ingress functions and 9 internal functions. There are 3 function chains in *Hello Retail!*. These functions also interact with other stateful back-end services and external services, *e.g.*, databases.

To evaluate the performance of DirectFaaS under different function chain lengths, we developed a serverless application with variant chain lengths. In our evaluation, we also use different internal delays between each node to simulate network conditions in different data centers.

### 5.1 Functionality Evaluation

DirectFaaS solves the challenges of releasing the API gateway from internal function invocations. Based on evaluations of real applications, we present the functionality of DirectFaaS.

*5.1.1 Authorization.* We deployed the network policy based on the invocation relationship of *Bookinfo* to restrict the internal function invocations. In each function, we invoke the vIP address of other functions at 10 requests per second (rps) for 30 seconds. The total requests are 300. As Table 1 shows, requests only successfully be forwarded to functions that have invocation relationships. Otherwise, requests will be blocked.

*5.1.2 Routing.* Since we use a specific vIP for each function, functions need to be able to respond when we invoke them with their corresponding vIP. We use an HTTP load generator (hey [43]) to simulate a burst of heavy traffic. We sent 10 concurrent requests to chain 3 of *Hello Retail!* for 120 seconds. The total requests to each function are 789. As Figure 8 shows, when we invoke the function chain using vIP, each internal function successfully responded to all requests, 789 in total.

*5.1.3 Resiliency.* As shown in Figure 8, when facing 789 requests, each function is scaled. Functions are scaled to different quantities of instances based on the number of invocations and resource consumption to handle requests. Requests are forwarded roughly evenly to each function. The reason the first function instance handles the most requests is that it lives the longest.

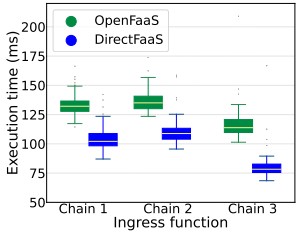 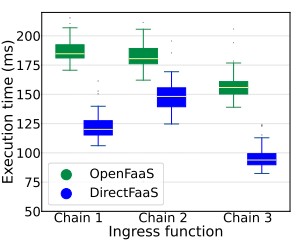 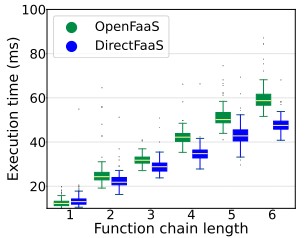 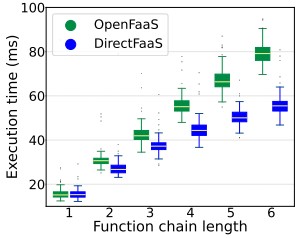

(a) Chain Invocation Time (0ms delay)  (b) Chain Invocation Time (0.5ms delay)  (c) Variant Chain Length (0ms delay)  (d) Variant Chain Length (0.5ms delay)

**Figure 9: Function chain execution time.** (a) The execution time of 3 function chains in *Hello Retail!* with 0ms delay. (b) The execution time of 3 function chains in *Hello Retail!* with 0.5ms delay. (c) The execution time of function chains of different lengths with 0ms delay. (d) The execution time of function chains of different lengths with 0.5ms delay.

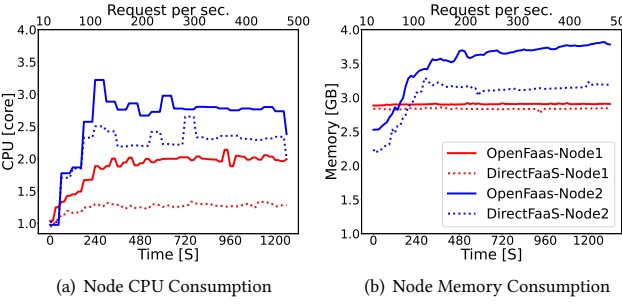

(a) Node CPU Consumption  (b) Node Memory Consumption

**Figure 10: Node Resource Consumption.** Heavy traffic bursts in *Hello Retail!*'s chain 3. The API gateway and DirectFaaS Controller are in the control node (Node1). The scaled function instances are in the worker node (Node2). On both the control node and worker node, DirectFaaS consumes fewer resources than OpenFaaS.

## 5.2 Runtime Performance

*5.2.1 Execution Time Reduction.* We use the time reduction of end-user response latency to demonstrate the advantages of Direct-FaaS. Specifically, we measure the time interval between an end-user sending a request and receiving the response from the serverless application. The latency reduction is also related to intra-cloud delays between nodes. Therefore, we measure scenarios of 0ms and 0.5ms internal delays. We compare DirectFaaS with baseline OpenFaaS. There are several research efforts [1, 19, 22, 25, 42, 51] dedicated to reducing serverless function invocation latency, but our work fundamentally different from them. The serverless network architecture has changed in DirectFaaS. We believe our work is complementary to them.

Figure 9 shows the response time of each function chain and the response time of variant function chain lengths over 100 repetitions. Figure 9(a) and 9(b) show the response time of each function chain in *Hello Retail!*. We measure the 0ms delay and the 0.5ms delay scenario. Averaging across 3 function chains in *Hello Retail!*, DirectFaaS reduces 24.4% and 30.9% invocation time with 0ms delay and 0.5ms delay, respectively. Figure 9(c) and 9(d) compare the invocation time of function chains of different lengths. The length of a function chain represents the number of functions in the chain. DirectFaaS can save up to about 24ms in one end-user request when the chain length is 6 with a 0.5ms delay.

*5.2.2 Resource Consumption Reduction.* We measure the resource consumption of DirectFaaS under varying loads. To do so, we make use of *Hello Retail!*'s chain 3 and use an HTTP load generator (hey [43]) to issue increasingly high external request loads ranging from 10 to 250 rps. Results for each load are reported over 120 seconds. The instances of functions will scale when the rps increases. The control components API gateway, and DirectFaaS Controller are deployed on the control node (Node1). Chain 3 of *Hello Retail!* is deployed on the worker node (Node2).

Figure 10(a) shows the per-node CPU. The control node CPU consumption grows at a gradual constant rate in OpenFaaS but grows slowly in DirectFaaS. It is because, with the external rps increases, the OpenFaaS API gateway not only needs to forward the external requests but also the internal requests. However, the DirectFaaS API gateway only needs to forward external requests. DirectFaaS consumes fewer CPU resources than OpenFaaS on the control node. DirectFaaS achieved a 30.1% reduction in CPU consumption on the control node. The worker node CPU consumption grows sharply at the beginning because of function instances auto-scaling. DirectFaaS consumes fewer CPU resources on the worker node because it scales fewer functions when receiving the same requests. DirectFaaS achieved a 15.4% reduction in CPU consumption on the worker node.

Memory consumption is reported in Figure 10(b). In the control node, OpenFaaS and DirectFaaS have similar memory consumption. Like the CPU consumption, the memory consumption on the worker node increases dramatically due to the increase in function instances. OpenFaaS consumes more memory than DirectFaaS because it scales more function instances. DirectFaaS reduces memory consumption by 13.8%.

To better show the response time of the function instance, we use the HTTP load generator to send 50 to 250 requests in 120 seconds. The function instance does not scale. As Figure 11 shows, DirectFaaS has less response time than OpenFaaS, meaning it can serve more rps. So under the same external rps, DirectFaaS only need fewer function instances to meet the requests.

*5.2.3 Overhead.* Compared with OpenFaaS, we introduce the DirectFaaS Controller to generate flow table rules when there are functions scaled. The DirectFaaS Controller will consume resources in the system. In Figure 12, we compare the control component of DirectFaaS with OpenFaaS. The OpenFaaS API gateway

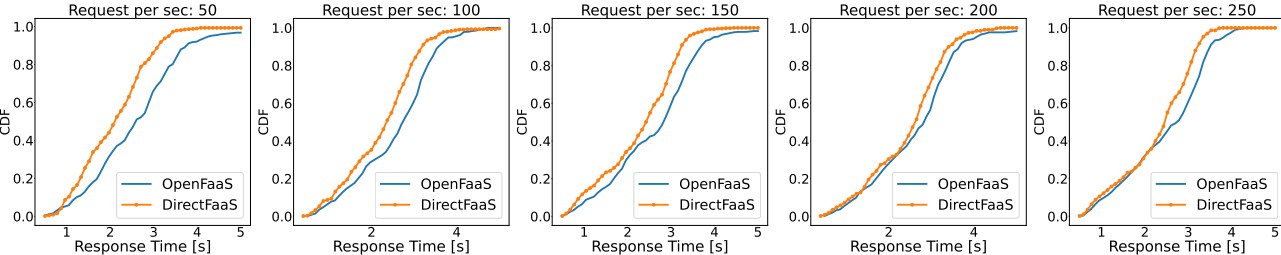

Figure 11: The product-purchase function chain response time at different requests per second. DirectFaaS has a smaller response time than OpenFaaS which can serve more requests per second.

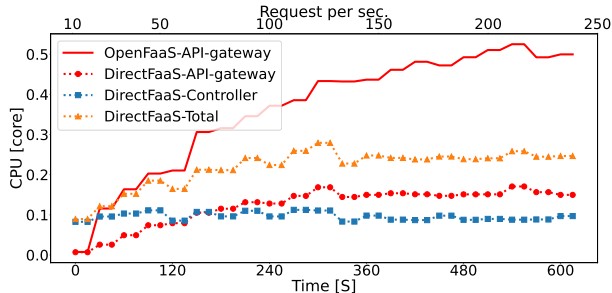

Figure 12: Component CPU Consumption. Heavy traffic bursts in *Hello Retail!*'s chain 3. OpenFaaS API gateway consumes more CPU resources than all of DirectFaaS control components.

consumes more CPU than the total control components of Direct-FaaS, including the DirectFaaS API gateway, and the DirectFaaS Controller. Although DirectFaaS Controller consumes additional CPU, the sum of their consumption is still less than the OpenFaaS gateway's CPU consumption.

There is a delay involved in generating the flow tables when functions are scaled. However, since flow tables are generated only during function scaling, they are concurrent with the function cold start time. The time to install a flow entry is in the millisecond range [44] and the time for the DirectFaaS Controller to generate a flow table is approximately 10 milliseconds, which is much smaller than the several seconds required for cold start times [34]. The flow table is deployed and ready by the time the function cold start completes, so it does not introduce any additional latency.

## 6 RELATED WORK

**Serverless Function Chain Latency.** Several research efforts focus on reducing latency for the serverless function chain. SPRIGHT [42] utilized shared memory communication to reduce the packet processing time on the worker node to lower latency. However, it only applies to functions on the same node. It can work with DirectFaaS to further reduce latency across worker nodes. QFaaS [22] emerges QUIC protocol to serverless platforms to accelerate function invocations while ensuring security. However, it still needs to establish a connection with the API gateway, with DirectFaaS, this connection can be reduced, further reducing latency. Boxer [54] uses a TCP hole-punching service in every function instance

to allow functions to communicate with each other. However, unlike DirectFaaS that makes no modification to function instances, Boxer adds services to the VM and will cause the overhead to light-weight containers which increases the start-up time. Xanadu [19] and Freshen [23] aim to reduce the latency of function chains by eliminating cascading cold starts, while DirectFaaS focus on reducing the latency between function communication. SAND [1], Nightcore [25], and Sequoia [51] focus on the function schedule sequence and schedule placement to reduce latency. However, they still follow the existing network architecture. DirectFaaS reduces function chain latency from a different angle of existing efforts and is complementary to them.

**SDN in Cloud Environments.** Software Defined Networks (SDNs) have been foundational in enabling virtualized networks for customer workloads in multi-tenant clouds. Antichi et al. [3] propose a full-stack SDN framework to alleviate the network management issues in the data center. Wang et al. [53] provide an SDN controller to each Infrastructure-as-a-Service (IaaS) cloud tenant to manage the network. But unlike DirectFaaS, this SDN controller is not associated with FaaS. With large cloud access traffic, Shao et al. [47] build a Disaggregated Software-defined Router (DSR) to keep up with the fast growth of traffic volume. Google proposes Orion [20], a distributed SDN platform to support system scalability. Inspired by these works, the design of DirectFaaS introduces SDN to serverless platforms, making it practical for removing the API gateway from internal function invocations.

## 7 CONCLUSION

DirectFaaS improves serverless function chain communication effectiveness. It reduces application execution time and resource consumption by removing the API gateway from the internal function invocations. With the creative use of SDN-based network management capability, DirerctFaaS achieves direct communication for serverless function chains. Compared to the current architecture using the centralized orchestrator to forward internal function invocations, DirectFaaS reduces 30.9% execution time when serving a complex web application. Additionally, when functions scale under high bursty concurrent requests, DirectFaaS reduces CPU consumption by 30.1% and memory consumption by 13.8% compared to OpenFaaS. DirectFaaS's code will be publicly available.

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
