# OpenReview forum: "DirectFaaS: A Clean-Slate Network Architecture for Efficient Serverless Chain Communications"
_ACM.org/TheWebConf/2024/Conference — TheWebConf24_

### Official Review · Reviewer_Ab6A · 2023-11-17

**Novelty:** 3
**Technical Quality:** 5

**Review:**

This paper proposes DirectFaaS a new network architecture to improve the communication in serverless computing platform. DirectFaaS decouple the control and data flows to reduce the overhead of API gateway. And it further uses virtual switch to implement required functions. Evaluation in  production serverless platform shows that DirectFaaS can effectively reduce the  application execution time.

Strengths：

-	By separating the control and data flows in serverless platform, DirectFaaS effectively reduce the overhead in the API gateway and improve the application execution time.

Major weaknesses:

-	The functions, operations and evaluation for the virtual switch are all not clear.

**Questions:**

I thinks this topic is interesting. However there are some limitations for this paper, and here are some comments to make.

1) OvS is a key function for the system. However this paper lacks description for it. For example, it is not clear how does the OvS update its rule by the API.  And I think evaluation for the time and overhead to update the rule in OvS can make this paper stronger.

2) The granularity of the authorization is not clear. This paper does not compare with granularity of authorization between the API gateway and the network policy. As the authorization by the network policy is IP level (function), whether it is enough for the system?

3) I really want to see some evaluation about how many rules are in the OvS when applying different applications in the serverless platform.  And how does the number of rules change when change the number of worker node. As the number of rules in the OvS is important which will effect its performance.

4) It is better to add some evaluation in the motivation to show how does the central architecture (API gateway) increase the latency. For example the latency between API gateway and worker node.

**Reviewer Confidence:**

3: The reviewer is confident but not certain that the evaluation is correct

**Scope:**

3: The work is somewhat relevant to the Web and to the track, and is of narrow interest to a sub-community

---

### Official Review · Reviewer_Hc2Z · 2023-11-23

**Novelty:** 5
**Technical Quality:** 5

**Review:**

This paper is about a clean-slate re-design of serverless chain communication. The approach is to separate control and data flow, enabling bypassing central gateway for the data communication. This is achieved by SDN-controlled forwarding. In addition, there is work on network policies for "multi-tenant traffic security."

The system has been integrated into the existing OpenFaaS platform and then evaluated with real-world server applications.

## Assessment

This paper addresses a relevant problem: the centralized design of serverless and microservice platforms. The presented approach is one way of addressing the problem and has been demonstrated to provide direct performance improvements. Overall, a nice paper.

I noticed a few problems:

1. Section 2.2., networking problems: P3 seems to be confusing traffic isolation with security. Even with flow isolation, you would need cryptography-based security to provide actual security.
2. Section 3.1, on the challenge about function chain authorization: yes, this could be address by a real authorization approach (crypto based).
3. Section 3.2, solution to C1: It's problematic to solve authorization by SDN forwarding. You should at least discuss the problem.
4. Section 6, Related Work: This seems to be missing the IETF SFC work. It should also discuss fully decentralized approaches, such Compute-First Networking in ICN.
5. The SDN approach is of course introducing potential central points of failure as well.

**Questions:**

1. Can you improve the discussion on security and traffic isolation as indicated above?
2. Can you include more related work, also from outside the directly related FaaS world?
3. Can you discuss potential issues of the SDN-based approach more?

**Ethics Review Description:**

no issues

**Reviewer Confidence:**

4: The reviewer is certain that the evaluation is correct and very familiar with the relevant literature

**Scope:**

3: The work is somewhat relevant to the Web and to the track, and is of narrow interest to a sub-community

---

### Official Review · Reviewer_GPV6 · 2023-11-23

**Novelty:** 5
**Technical Quality:** 6

**Review:**

Thank you for your submission. This work presents DirectFAAS, a mechanism for improving server-less function chain performance by eliminating the need for intermediate communication between functions and an API gateway. In particular, the authors take advantage of an SDN-based setup, which separates the data and control planes, allowing the programmable network to handle many of the API gateway tasks (such as function authentication), enabling functions to communicate directly, reducing communication overhead.

The arguments presented are clear and well motivated, with the technical descriptions easy to follow.

Overall, DirectFAAS presents a clever application of SDNs to the problem of function chaining and the evaluations show it is successfully able to achieve the required behaviors with less system resource utilization.

My primary concerns here come from the evaluation, which only compares to the OpenFaaS baseline. While showing improvement over such cases, I think it important to understand how the components of directFaaS compare to other systems — which may require potentially less rearchitecture to the system  (though as the authors note less changes to the functions themselves). While the authors further note that existing techniques are complementary, the comparisons would still be valid.

Some components of the evaluation were also relatively light on details of validation for why certain load conditions were selected. Further details would strengthen the nature of those arguments (see detailed comments for specifics).

Finally, in the evaluation, it is noted that some of the improvements observed, especially for the load on the compute nodes, comes from the fact that DirectFaaS is able to scale to fewer instances — however it is not clear why this would be the case, since the data collected by the virtual switches and fed to the scaler are likely to be similar to that observed by the API gateway itself.

Pros:
- Clear motivation, presentation, and description of the system
- Provides a clever application of SDNs to help streamline function chains
- Evaluation demonstrates improvements in tested scenarios

Cons:
- Evaluation fails to compare to any other mechanisms besides the OpenFAAS baseline.

Detailed Comments:

- 2 - It seems like some of these problems could be solved without broad rearchitecting, including distributing some tasks, or implementing limited buses or similar. It's not clear to me that would be all that important, but is a possibility to consider.

- 3.2 -  The virtual network enforcing access control ultimately seems to offer a more constrained set of access controls, e.g. a binary determination of whether or not two functions can communicate, but it seems like the API gateway can likely implement arbitrarily complex access control.

- 5 - would like some more detail about the home made-chain length application, and the corresponding delay values. Are these simple artificial delays between functions?

- 5.1.1 - Im not sure this component of the evaluation offers much — this section feels tautalogical.

- 5.1.2 - Routing load test — 10 concurrent requests seems low, but I have no frame of reference. Why 10? Why not 100? 1000?

- 5.2.1 points to other techniques being developed to improve performance — even in the case of complementary designs, it would be valuable to consider their relative performance, as it allows the reader to assess how much performance has been gained.

- 5.2.1 - Figure 9 whats the delay here? Delay between functions? Is this on an otherwise unloaded system?

- 5.2.2 - The numbers in the text dont appear to match the figures (10-250rps vs 10 - 500rps, 120s vs 1200s)

- Why does DirectFaas scale fewer instances? Shouldn't it be approximately the same, with the differences primarily being in the overhead paid with function costs? (Up to the api which might scale more, but that should be on the control node?)

- There is a note that faster response time means it can serve more RPS — isn't this likely to depend on the specific resources requirements of the requests?

Nits:
- C2: "Functions scales a different number of instances based on the traffic volume. " -  is this sentence out of place?

 4 "starts" -> "stars"

**Questions:**

How does DirectFaaS compare to other (albeit Non-SDN) mechanisms?

What is the primary reason that DirectFaaS is able to use fewer resources (esp in Figure 10).

**Reviewer Confidence:**

3: The reviewer is confident but not certain that the evaluation is correct

**Scope:**

4: The work is relevant to the Web and to the track, and is of broad interest to the community

---

### Official Review · Reviewer_xLmz · 2023-11-26

**Novelty:** 6
**Technical Quality:** 5

**Review:**

In this paper, the authors address the challenge of cloud-based network chaining. Starting from the observation that current deployments rely on a dedicated API gateway, the authors introduce a peer-to-peer model to allow direct interaction between remote functions. The proposed solution is inspired by SDN.

The paper is well written and the thoughts are clear. My main problem is the assumption that the API gateway does not scale and becomes a bottleneck. Even though the authors show that their approach outperforms an API gateway it remains unclear whether an API gateway is a bottleneck in practice. For example, the authors argue that an API gateway becomes a bottleneck for traffic (page 5, line 541) but you could also argue that the traffic is rather low.

**Questions:**

Can you clearly show that the API gateway is a bottleneck in practice?

Can you explain the overhead introduced by the SDN approach? Which additional costs (resources, budget ...) arise by introducing DirectFaas?

**Reviewer Confidence:**

3: The reviewer is confident but not certain that the evaluation is correct

**Scope:**

3: The work is somewhat relevant to the Web and to the track, and is of narrow interest to a sub-community

---

### Official Review · Reviewer_Gn1n · 2023-12-04

**Novelty:** 2
**Technical Quality:** 4

**Review:**

Paper Summary:
The main bottleneck is the latency caused by the centralized API gateway. The authors argue that we should establish a direct secure channel between functions to remove the bottleneck through a novel network architecture. The authors design a system to support direct invocation mechanisms using SDN and improve security with fine-grained network policies. The experiments show that the proposed platform DirectFaaS can achieve greater performance.

 Strengths:
The design of the system is reasonable and clear. This work will be open source.


Weakness:
The novelty is not strong as existing serverless platforms do support chaining functions with the direct invocation mechanism. The evaluation misses critical related works.

Additional Comments:

My main concern is that the main benefits to justify proposed network architecture are not well presented, as removing the centralized bottleneck can increase performance, or fine-grained networking policy improves performance is well-understood.  Current serverless platforms such as AWS lambda already support direct calling of subsequent functions. Besides, the nightcore paper that this paper cites has already addressed the issues of internal function call bypassing API gateway. Even for the cases when function consolidation on a single server is not possible, Nightcore’s engines can architecturally forward requests directly among the worker servers. I encourage the authors to dive deeper with newer use cases for more novel design motivations.

**Questions:**

What specific threats or vulnerabilities do the new network architecture address compared to API-gateway methods?

**Reviewer Confidence:**

4: The reviewer is certain that the evaluation is correct and very familiar with the relevant literature

**Scope:**

3: The work is somewhat relevant to the Web and to the track, and is of narrow interest to a sub-community

---

### Decision · Program_Chairs · 2024-01-22

**Decision:**

Accept

**Comment:**

The authors propose a design for bypassing API gateways in serverless function invocations between function instances of a chain to improve performance. to this end, they design an SDN-based solution.

 Those reviewers who commented on the authors rebuttals appear to be happy with the responses they received. There were, however, two main aspects that may not have been addressed completely:

 1. Security. While, as the authors point out, their paper does not focus on security, using IP layer mechanisms to ascertain who is authorised to invoke a function may not be sufficient. The discussion lacks the necessary details but, given the trend to zero trust security in which all operations need to be authorised, it is unclear if and how the necessary security properties can be upheld past the API gateway. This is not about external traffic but about traffic from different function instances.

 2. The word "clean slate" is a grand one and often used in the past to refer to completely new network architectures. This paper clearly doesn't aim as high but the response to one reviewer comment on comparison to SFC and CFN is not convincing. Both approaches seem to be similar in nature in optimizing traffic flows and allowing for direct interaction between successive functions without detours through a gateway.

 I'd personally be curious on the implications of reach of function calls for function chaining. It would appear to be that applying SDN will confine function spread to a single DC while other mechanisms might be more flexible. But this is maybe something left for discussions or future work considerations.

 Overall, while some technical scores are different (even more so concerning novelty), the reviewers seem to largely agree on the merit of the paper, so this makes it a weak accept.